# Genetic Concordance in Primary Cutaneous Melanoma and Matched Metastasis: A Systematic Review and Meta-Analysis

**DOI:** 10.3390/ijms242216281

**Published:** 2023-11-14

**Authors:** Thamila Kerkour, Catherine Zhou, Loes Hollestein, Antien Mooyaart

**Affiliations:** 1Department of Dermatology, Erasmus MC Cancer Institute, 3015 GD Rotterdam, The Netherlands; t.kerkour@erasmusmc.nl (T.K.); c.zhou@erasmusmc.nl (C.Z.); l.hollestein@erasmusmc.nl (L.H.); 2Department of Pathology, Erasmus MC Cancer Institute, 3015 GD Rotterdam, The Netherlands

**Keywords:** concordance, primary cutaneous melanoma, metastasis, *BRAF*, *NRAS*, *c*-*KIT*

## Abstract

Studying primary melanoma and its corresponding metastasis has twofold benefits. Firstly, to better understand tumor biology, and secondly, to determine which sample should be examined in assessing drug targets. This study systematically analyzed all the literature on primary melanoma and its matched metastasis. Following PRISMA guidelines, we searched multiple medical databases for relevant publications from January 2000 to December 2022, assessed the quality of the primary-level studies using the QUIPS tool, and summarized the concordance rate of the most reported genes using the random-effects model. Finally, we evaluated the inter-study heterogeneity using the subgroup analysis. Thirty-one studies investigated the concordance of *BRAF* and *NRAS* in 1220 and 629 patients, respectively. The pooled concordance rate was 89.4% [95% CI: 84.5; 93.5] for *BRAF* and 97.8% [95% CI: 95.8; 99.4] for *NRAS*. When high-quality studies were considered, only *BRAF* mutation status consistency increased. Five studies reported the concordance status of c*-KIT* (93%, 44 patients) and *TERT* promoter (64%, 53 patients). Lastly, three studies analyzed the concordance of cancer genes involved in the signaling pathways, apoptosis, and proliferation, such as *CDKN2A* (25%, four patients), *TP53* (44%, nine patients), and *PIK3CA* (20%, five patients). Our study found that the concordance of known drug targets (mainly *BRAF*) during melanoma progression is higher than in previous meta-analyses, likely due to advances in molecular techniques. Furthermore, significant heterogeneity exists in the genes involved in the melanoma genetic makeup; although our results are based on small patient samples, more research is necessary for validation.

## 1. Introduction

Cutaneous melanoma (CM) is an aggressive tumor arising from the pigment-producing cells (melanocytes) located in the skin. The incidence of melanoma has steadily increased in many susceptible populations over the past five decades [1]. Patients diagnosed with localized disease undergo surgical resection with a favorable prognosis [2]. However, 40% of those patients develop metastatic disease, reducing the 5-year overall survival to less than 30% [3,4]. Unraveling the molecular biology of CM has revolutionized treatment with targeted therapy (BRAF and MEK inhibitors) and immunotherapy (anti-PD1 and anti-CTLA-4 agents) [5,6,7,8,9,10]. Patients with metastatic tumors undergo molecular testing for the *BRAF* V600 mutation to determine treatment options with BRAF/MEK inhibitors [7,11,12]. With the discovery of additional melanoma driver genes, strong efforts were made to develop targeted therapies for BRAF wild-type tumors [13]. Several researchers investigated potential targeted treatments for *NRAS* Q61 mutant melanomas. Ongoing clinical trials tested the new treatments with MEK and CD147i inhibitors in advanced melanoma after immunotherapy failure [14,15]. Moreover, further inhibitors are also available for melanoma harboring an amplified *c-KIT* gene [16,17]. The mutations in *BRAF* V600, *NRAS* Q61, and *c-KIT* occur early in melanoma development [18,19]. Consequently, if a patient develops a metastasis, the absence of these alterations in the primary tumor may be sufficient to exclude the patient from the targeted therapies. Otherwise, an invasive biopsy of the metastatic deposit needs to be performed. However, a meta-analysis conducted in 2017 showed an overall discrepancy rate of 13.4% of the *BRAF* status between the primary melanoma and the matched metastasis [20]. Hence, it is advised to determine the *BRAF* status in the metastasis deposit [21].

With advances in sequencing techniques for mutation detection, additional studies addressed the concordance of the mutation status of primary CM and matched metastasis in multiple cancer genes. However, the sensitivity of these methods for mutation detection has not yet been compared to older techniques. Therefore, collecting up-to-date information on the concordance status of *BRAF* and additional driver genes, along with assessing the quality of the new studies, may help to redirect the diagnosis and therapeutic decisions for CM patients. Additionally, understanding how CM tumors evolve and differ between primary and metastatic sites is important. Thus, in this study, we aimed to systematically review all the literature and undertake the meta-analysis where appropriate in order to determine the mutation concordance rate between primary CM and its matched metastatic sites.

## 2. Methods

We conducted this systematic review and meta-analysis in accordance with the Preferred Reporting Items for Systematic Review and Meta-Analysis (PRISMA) guidelines. The preregistered protocol is available at PROSPERO under the protocol number CRD42022327641, accessed on 15 May 2022 and available from: https://www.crd.york.ac.uk/prospero/display_record.php?ID=CRD42022327641. 

### 2.1. Search Strategy and Eligibility Criteria

A comprehensive search strategy was employed to ensure the inclusion of relevant studies. We performed an initial search in Embase to detect the relevant keywords in the titles and the abstracts. The primary search terms used in Embase were “genetic features”, “primary cutaneous melanoma” and “metastatic melanoma”. The obtained synonymous terms from Embase were then added to enhance the search strategy. Articles published from 2000 to December 2022 were searched in the following databases: Medline, Embase, Web of Science, and the Cochrane Central Register of Controlled Trials. The search strategies are available in Appendix A. Two independent reviewers (C.Z. and T.K.) evaluated the titles and abstracts of the identified studies in separate EndNote libraries. Discrepancies were discussed with a third reviewer (A.M.). Inclusion criteria consisted of studies that compared the genetic patterns between primary CM and the matched metastasis within the same patient. Studies that only included uveal and mucosal melanoma were excluded, as well as conference abstracts and case studies. 

### 2.2. Data Extraction and Quality Assessment

Full-text screening of all the relevant articles was conducted by two reviewers (C.Z. and T.K.). Data extraction was independently performed by the two reviewers (C.Z. and T.K.) using a customized data extraction Excel sheet. The following information was extracted from each study: The first author’s name and their affiliated country, the year of the publication, the technique used to determine the mutation status, the analyzed genes, the total number of patients, and the number of patients with concordant status for each gene. To assess the quality of each included study, one reviewer (T.K.) used the Quality in Prognosis Studies (QUIPS) tool [22] and discussed the outcomes with a second reviewer (A.M.) to ensure consensus. This tool consists of six domains: Study participation, study attrition, prognosis actor measurement, study confounding, statistical analysis, and reporting. For each domain, the score was assigned: High risk of bias = 0, moderate risk of bias = 0.5, and low risk of bias = 1. Studies with a total score of ≥4 were considered high quality, whereas studies with a total score of <4 were considered low quality.

### 2.3. Outcome of Interest

The outcome of interest in the meta-analysis is defined as the concordance rate of the mutation status between the primary cutaneous melanoma and their paired distant metastasis in each single gene. For this study, we considered meta-analysis only for the genes that were reported in more than ten studies. Therefore, we conducted the meta-analysis for *BRAF* V600 and *NRAS* Q61 mutations. We reported the available data regarding additional genes in a systematic manner. 

### 2.4. Pathway Analysis

To determine if the mutations are associated with pathways that may be involved in the metastasis progression, we used the web-free server g:profiler [23] to identify the possible pathways associated with the mutated genes that we mutated in both primary and metastasis. We selected the top three pathways with the highest *p*-value. g:profiler output list details are presented in Appendix A.

### 2.5. Statistical Analysis

Differences in the concordance of *BRAF* status between male and female patients were assessed using a Student’s *t*-test. Random-effects meta-analysis models were used to pool the concordance rate and the confidence interval (CI) across all the studies for each gene (*BRAF* and *NRAS*) because combining all the genes may be biased. Freeman–Tukey double arcsine transformation was applied to weigh the individual studies [24]. To assess the heterogeneity across the studies, we performed a chi-square test to estimate the I^2^ statistic (0–100%, 0% indicated no heterogeneity).

To explain potential sources of heterogeneity, pre-specified potential effect modifiers, such as the technique of detecting the mutation (molecular vs. immunohistochemistry) and the study quality (QUIPS total score), were considered. Subgroup analyses were used to calculate the concordance rates for each group. Small study effects and publication bias were assessed using funnel plots and Egger’s test (Appendix A). The analyses were performed using the meta [25] and the metaphor packages [26]. All the analyses were done in R version 4.2.13.

## 3. Results

### 3.1. Search Results and Studies Characteristics

The PRISMA flowchart of the selection process is presented in Figure 1. A total of 2466 studies (after excluding duplicates) were identified from the selected databases. Screening and evaluation of eligibility criteria resulted in 40 studies being considered for qualitative assessment, with 31 included in the quantitative assessment. The majority of studies included in the meta-analysis (25) focused mainly on *BRAF* concordance status. For the studies that included mucosal or uveal melanomas, we only extrapolated the mutation data of patients with CM.

### 3.2. The Genetic Concordance between Primary Cutaneous Melanoma and Matched Distant Metastasis

The concordance in patients with cutaneous melanoma was reported mainly in the melanoma driver genes. Twenty-five studies [27,28,29,30,31,32,33,34,35,36,37,38,39,40,41,42,43,44,45,46,47,48,49,50,51] compared the mutation status of *BRAF* with the median concordance rate of 88% (range 56–100, total patients = 1220) (Table 1). *BRAF* was mainly reported as either muted in position V600 or wild type. *NRAS* status was the second most highly reported gene in 14 studies [28,30,32,38,43,45,48,49,50,51,52,53,54,55] (Table 2), with a median concordance of 97% (range 85–100, total patients = 629), and the mutation site was mainly focused on the protein position Q61. *c-KIT* concordance was reported in 3 studies [32,38,49] with a median concordance rate of 100% (range 88–100, total patients = 44) (Table 3). Finally, the concordance of the *TERT* promoter was reported in two studies [45,50], and the media concordance was 64% (range 55–68, total patients = 53) (Table 4).

Three studies [43,48,49] compared the mutational status between primary melanoma and their matched metastasis tumors in multiple cancer genes. Details of each gene are described in Table 4. Most of the reported mutated genes were reported in both the primary and metastasis tissues, except *KRAS* G12A and *NEK E379K* mutations that were found in primary tumors and *CCND1* G103R in the metastasis tumor. To understand the role of these mutations, we looked at their possible association with pathways known in cancer progression. Most of the genes are involved in proliferation, cell cycle, and cellular senescence. The details of the pathways analysis are reported in Appendix A.

### 3.3. The Genetic Concordance between Primary Cutaneous Melanoma and Matched Distant Metastasis According to Metastatic Site and Gender

To determine the impact of the metastatic site and gender on the concordance rate between primary melanoma and metastatic deposit, we extracted data on the tissue type of the metastasis (Table 5) and patient gender (Table 6) from each study. However, due to the unavailability of data for *NRAS* and *KIT*, we could only report the data for *BRAF*.

The lymph node (excluding sentinel lymph node) was the most commonly reported metastatic site, with eight studies encompassing 262 patients. The skin was the second most frequently reported site, with six studies comprising 84 patients. Other metastatic sites, including visceral, brain, and subcutaneous locations, were reported in limited studies, with 50, 29, and 18 patients, respectively. The concordance rates for *BRAF* according to gender were reported in 5 studies. Although males represented a slightly larger proportion, no statistically significant difference in the concordance rate between females and males was observed (*p*-value = 0.19).

### 3.4. Meta-Analysis of the Concordance Rate for BRAF and NRAS

We assessed the concordance rate for *BRAF* in 1220 patients and for *NRAS* in 629 patients (Figure 2 and Figure 3). We calculated the concordance rate separately for wild-type versus mutated status for each gene. Because we focused on cutaneous melanoma in this study, we excluded uveal and mucosal melanoma patients in five studies [23,25,28,35,38].

The pooled random effects concordance rate for *BRAF* was 89.4% [95% CI: 84.5; 93.5] and 97.8% [95% CI: 95.8; 99.4] for *NRAS*. The heterogeneity between studies was high for the *BRAF* concordance rate (I^2^ = 74%, *p* < 0.01). However, for *NRAS*, the risk of bias was low (I^2^ = 7%, *p* = 0.38).

### 3.5. Subgroup Analysis According to the Technique and QUIPS Tool Score

The mutation status of *BRAF* and *NRAS* in the reported studies may be affected by the study design and the techniques used for mutation detection. For *BRAF* detection, 16 studies utilized molecular-based techniques, two utilized IHC, and seven used both. The pooled *BRAF* concordance rate was 86.4% [95% CI 79.6; 92.1] with a molecular-based technique, 92.8% [95% CI: 86.3; 97.5] with IHC, 81.9% [95% CI: 82.2; 98.4] with IHC, and a molecular-based technique combined. However, these differences were statistically insignificant (*p*-value = 0.24) (Figure 4). We did not perform subgroup analyses based on detection techniques on *NRAS* concordance because most of the studies used molecular-based techniques.

Other confounders, such as sample collection, analysis interpretation, and the type of statistical tests in the individual studies, may affect the mutation detections and, subsequently, the concordance rate. Therefore, we performed a subgroup analysis based on the QUIPS score for *BRAF* and *NRAS* concordance rates (Figure 5 and Figure 6). The pooled concordance rate for BRAF was 91.4% [95% CI: 86.7; 95.3] for studies with a high QUIPS score (≥4) and 81.0 [95% CI: 68.1; 91.6] for studies with a low QUIPS score (<4), (*p*-value = 0.03). The pooled concordance rate for NRAS was 95.7% [95% CI: 89.7; 99.5] for studies with a high QUIPS score (≥4) and 99.1% [95% CI: 97.3; 100] for studies with a low QUIPS score (<4). This difference was statistically insignificant (*p*-value = 0.38).

## 4. Discussion

In this study, we comprehensively analyzed the current knowledge on the concordance of mutated genes between primary CM and their matched metastasis. Most of the reviewed studies focused on the concordance of *BRAF* mutation status because it has clinical value in the treatment decision for BRAF inhibitors. The second-most studied gene was *NRAS.* The overall pooled concordance rates for *BRAF* and *NRAS* status were high (88.8% and 97.2%, respectively). However, the reported concordance rates in the individual studies varied widely, ranging from 56% to 100% for *BRAF* and 85% to 100% for *NRAS*. Among the factors that could explain this wide range are: Mutation detection technique, gender, metastatic site type, and study quality. We observed that all these factors were of influence on the concordance rate. First, we explored the impact of mutation detection techniques. For *BRAF*, the concordance rate was higher when an IHC-based technique was used (92.8%) and lower when molecular-based techniques were used (86.4%). Although we could not explore the effects of gender and metastatic sites with significant power, we did see for *BRAF* concordance rate the most discordant cases in the direction of female sex and skin as a metastatic site.

Only five studies [43,45,48,49,50], with study sizes ranging from 3 to 41 participants, analyzed the variability in the mutation profiles using panel cancer driver genes. The alterations in *TERT*, *CDKN2A*, *TP53*, *RPPIK3CA*, and *EPHB6* were simultaneously detected in matched primary and metastasis tumors, albeit in only a small subset of patients. In contrast, other mutated genes were found exclusively in either primary or metastatic tumors. Predominantly, the majority of the genes displaying alterations were associated with critical pathways involved in tumor invasion, including those regulating proliferation, cell cycle, and apoptosis. This is in line with the current knowledge on melanoma progression [56,57,58]. In essence, investigating the sequential acquisition of mutations will be required to fully identify the key driver mutations that are responsible for the processes of invasion, adaptation, and, ultimately, melanoma metastasis dissemination [59]. 

CM presents cancer with a very high tumor mutational burden (TMB) [60]. This characteristic may lead to polyclonal formation within the individual primary melanoma tumor. In other words, a primary tumor can comprise a mixture of cells presenting different genetic profiles of the same gene, with the ability to metastasize to other tissue organs [61]. Two studies showed intratumoral heterogeneity within the same melanoma samples for *BRAF* and *TERT* mutation status using different micro-dissected regions of the same sample [29,50]. A meta-analysis reported similar observations when investigating *EGFR* and *KRAS* mutation status in matched primary tumors and distant metastasis of non-small cell lung cancer, which is also considered a tumor with a TMB [62]. Thus, intratumoral heterogeneity may be a common feature of high-TMB tumors. In our study, we also showed that additional mutations in genes such as *KRAS* and *NEK10* were likely to be seen in primary melanoma. In contrast, mutations in genes such as *TERT* and *CCND1* were likely present in metastasis. Thus, we highlight the need to investigate the clonality of more melanoma-specific genes, especially in the case of building metastasis risk prediction models that are based on the genetic profile of primary tumors, as the primary tumors may contain driver genes that are irrelevant for the recurrence of the metastasis. For example, Chang et al. have shown that within the same melanoma patients, the primary tumor had a *BRAF* mutant status, but the metastatic tumor had lost this mutation and acquired a new mutation in the *TERT* promoter [50]. As a result, a metastasis risk prediction model based on the *BRAF* mutation status of the primary tumor would not be accurate for this patient. Therefore, it is essential to develop metastasis risk prediction models that take into account the intratumoral heterogeneity of melanoma tumors. This could be done by sequencing multiple regions of the primary tumor or by sequencing both the primary and metastatic tumor tissues.

The accuracy of the mutation detection technique may have an effect when comparing the mutation status [63,64]. Bruno et al. tested the concordance of *BRAF* status between 25 paired primary and matched melanoma samples by applying both IHC and real-time PCR. Their study showed that IHC was more effective in detecting the signal of mutated *BRAF* V600E [41]. This finding suggested that IHC is the stable method for the *BRAF* testing method, especially for detecting the most common *BRAF* mutation, V600E. From a molecular point of view, IHC is subject to some limitations. For instance, IHC can be subjective as the pathologist’s interpretation affects the results. IHC may not detect mutations in all cases, particularly when the tumor sample is small or if the mutation is present in only a small percentage of tumor cells [65]. At the same time, molecular-based techniques such as real-time PCR are more objective and sensitive than IHC [66]. In the daily practice of *BRAF* testing, molecular techniques are considered the golden standard [67]. 

The tumors can evolve over time, and new mutations can arise in the metastases. Thus, the time to metastasis, which is defined as the time between the primary tumor and the metastasis tumor biopsy, may also affect the mutation concordance rate. In our study, we were unable to analyze the impact of time on metastasis as there were not enough eligible studies. It is essential to mention that no new studies have reported more extensive information on the timing of metastasis since the previous meta-analysis of *BRAF* [20]. In colorectal cancer, the concordance rate of *KRAS* status decreases when the time to metastasis increases [68,69]. Thus, it is recommended to perform more frequent surveillance for patients with a longer time to metastasis, as they may be at higher risk for developing metastases with a different KRAS mutation status [68].

The metastatic site plays a crucial role in changing the mutation status between matched primary and metastasis melanoma tumors. Firstly, the microenvironment of the metastasis site can exert a unique selection pressure on the melanoma cancer site. In other words, the metastasis tissue site can offer distinct biochemical, immune, and structural conditions that differ from the skin site. This means the melanoma cells can adapt and acquire new mutations to thrive in the specific microenvironment of the metastasis site. For example, the melanoma cells that metastasize to the brain may develop mutations that evade the immune system, as the brain is an immune-privileged site [70]. Melanoma that metastases to the lung may develop mutations that allow it to resist the effects of chemotherapy, as the lung is a vascularized organ [71]. 

The overall study quality had no significant effect on the concordance rates of *NRAS* mutation status. Yielding that the reliability of *NRAS* mutation status determination was fairly consistent across studies, regardless of their quality ratings. However, we observed higher concordance rates for *BRAF* mutation status in studies rated as high quality compared to studies rated as low quality. These findings suggest that beyond mutation detection techniques, other factors such as patient selection and time and type of fixation, as well as improper tumor sampling, may affect the concordance rate of the mutation profile [72]. From a biological standpoint, it is worth noting that both *BRAF* and *NRAS* mutations are present in nevi, suggesting that such mutations might represent early events that could already be present in all tumor cells. This is also supported by the results seen in immunohistochemistry (IHC) [29,73]. It appears that, based on the available data, discrepancies in the current literature may be attributed to sample management and mutation detection techniques rather than being solely due to tumor heterogeneity. Refining and standardizing sample collection and analysis procedures are crucial to enhancing the accuracy and consistency of mutation profiling in melanoma cancer research. Ultimately, this leads to more reliable insights into this complex disease.

Our study has several limitations; firstly, the original studies that we included in our meta-analysis for *BRAF* and *NRAS* status were relatively small, which limited the power of our study to detect statistically significant results. Secondly, most of the included studies used different criteria to select patients, mainly for the metastasis tumor (lymph node or distant metastasis). Lastly, we did not have access to clinical data for all of the patients in the original studies, which prevented us from analyzing the effect of clinical variables, such as the tumor thickness and the presence of the ulceration, on the concordance rate. Despite these limitations, our study presents the first meta-analysis that pooled the *NRAS* concordance rate in matched primary and metastasis of cutaneous melanoma. This is an important finding, as NRAS mutations are responsible for a significant proportion of melanomas, but no well-established NRAS-targeted therapies are yet known. In addition to the updated data about the pooled *BRAF* concordance rate from the previous meta-analysis. 

Our results consolidate the evidence of the mutation concordance between primary melanoma and matched metastasis. We demonstrated a high concordance for *BRAF* and *NRAS* mutation status. We observed that the *BRAF* concordance rate is higher than reported in the last systematic review. This finding may suggest that advances have been made in molecular techniques and IHC techniques. Nevertheless, further investigations are required to comprehend the intricacies of melanoma, including its genetic heterogeneity and the complex interaction of multiple genes driving the metastatic progression. Expanding the scope of our study is essential to attaining a comprehensive understanding of tumor heterogeneity and the genetic changes that take place throughout the metastatic evolution of melanoma. This should include not only a wider range of genes but also the integration of more clinical information, as these multi-faceted forms of research will aid in uncovering the complexities of the genetic landscape of melanoma.

## 5. Conclusions

The complete explanation of the genomic background behind melanoma pathogenesis remains elusive. This study presents a summary of current understanding regarding the clonal relatedness of mutated genes in primary cutaneous melanoma and matched tumors. We draw attention to the knowledge gap regarding the genetic heterogeneity of melanoma-specific genes, which could serve as a valuable tool in identifying suitable patients for trials testing possible new targeted therapies. Furthermore, we found that CM presents a significant concordance in *BRAF* status between primary and metastatic tumors than in previous meta-analyses, probably due to technical advances, although a minority of patients showed inconsistencies. Therefore, we propose that molecular testing in metastatic tissue is a more appropriate method for determining *BRAF* status to tailor the treatment decision, although it is still reasonable to use the primary tumor in cases of difficulty. Further research should investigate the factors causing these discrepancies and the feasibility of using other molecular markers with *BRAF* status to better stratify patients for melanoma treatment.

## Figures and Tables

**Figure 1 ijms-24-16281-f001:**
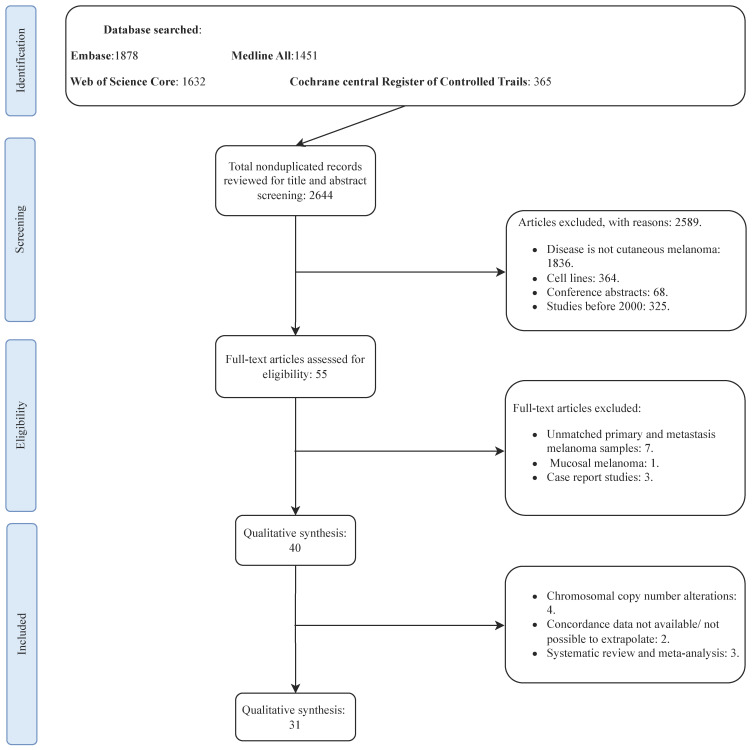
PRISMA flow diagram of the literature search.

**Figure 2 ijms-24-16281-f002:**
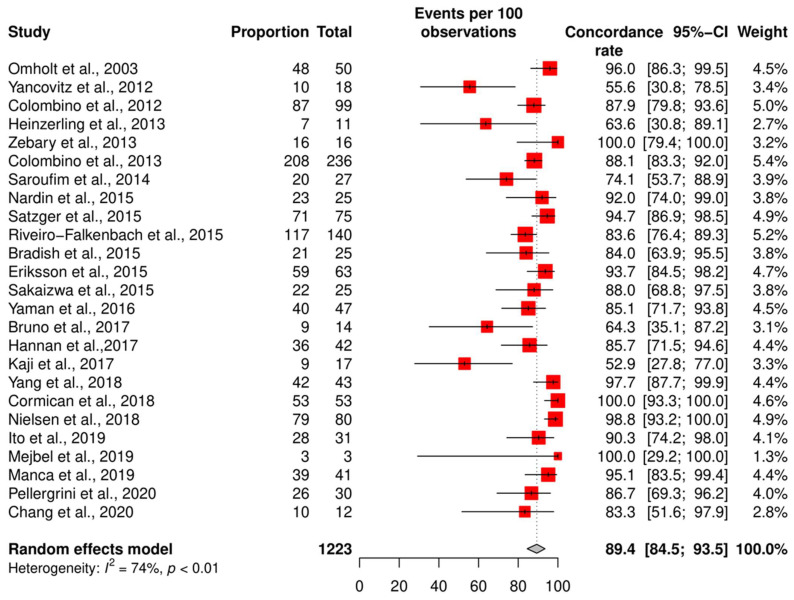
Results of the *BRAF* meta-analysis. Twenty-five studies were included to pool the concordance rate of *BRAF* status [27,28,29,30,31,32,33,34,35,36,37,38,39,40,41,42,43,44,45,46,47,48,49,50,51], red square: point estimate of each study, and grey diamond: summary estimate of the total studies.

**Figure 3 ijms-24-16281-f003:**
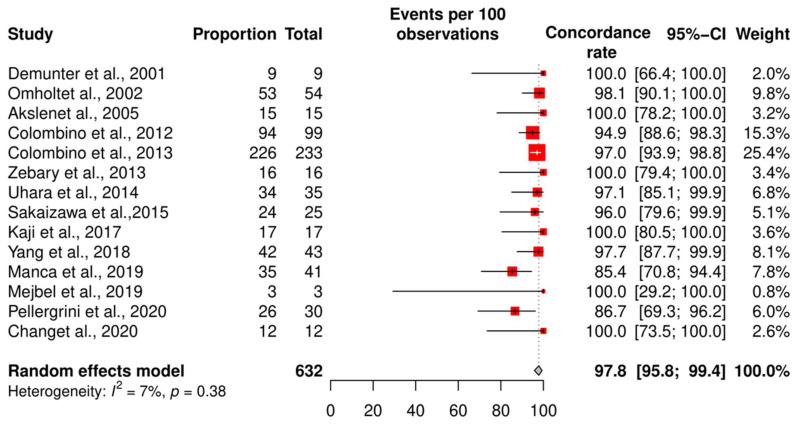
Results of the meta-analysis *NRAS.* Fourteen studies were included to pool the concordance rate of *NRAS* status [28,30,32,38,43,45,48,49,50,51,52,53,54,55], red square: point estimate of each study, and grey diamond: summary estimate of the total studies.

**Figure 4 ijms-24-16281-f004:**
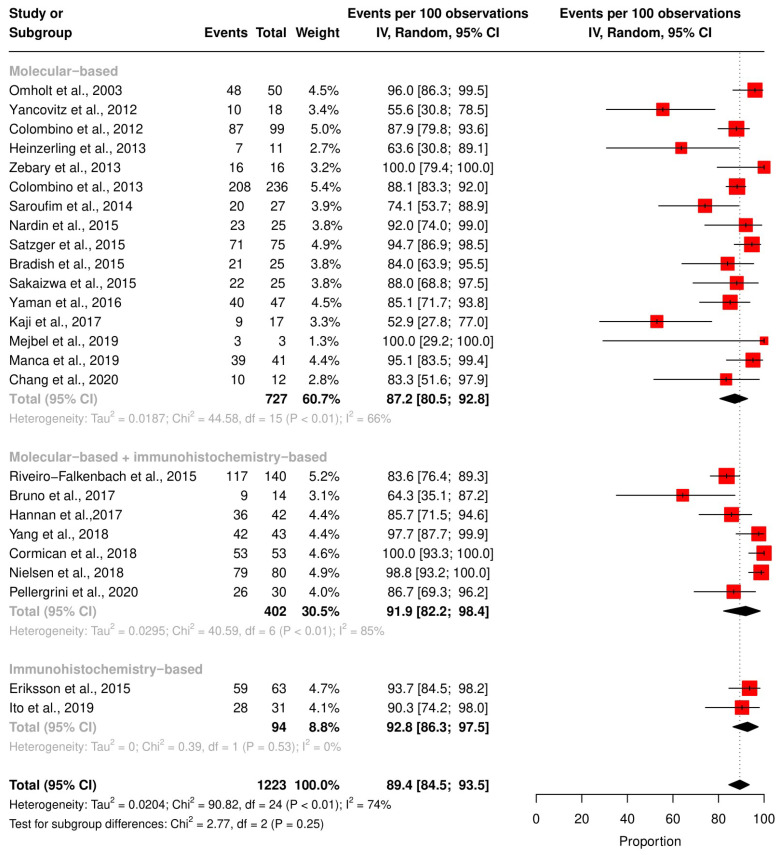
Subgroup analysis for *BRAF* concordance rate based on mutation detection methods. Studies with molecular-based technique only [27,28,29,30,31,32,33,34,36,38,39,40,43,48,49,50], studies with molecular-based + immunohistochemistry-based technique [37,41,42,44,45,46,51], studies with immunohistochemistry-based technique only [35,47], red square: point estimate of each study, and black rhomb: summary estimate of each subgroup.

**Figure 5 ijms-24-16281-f005:**
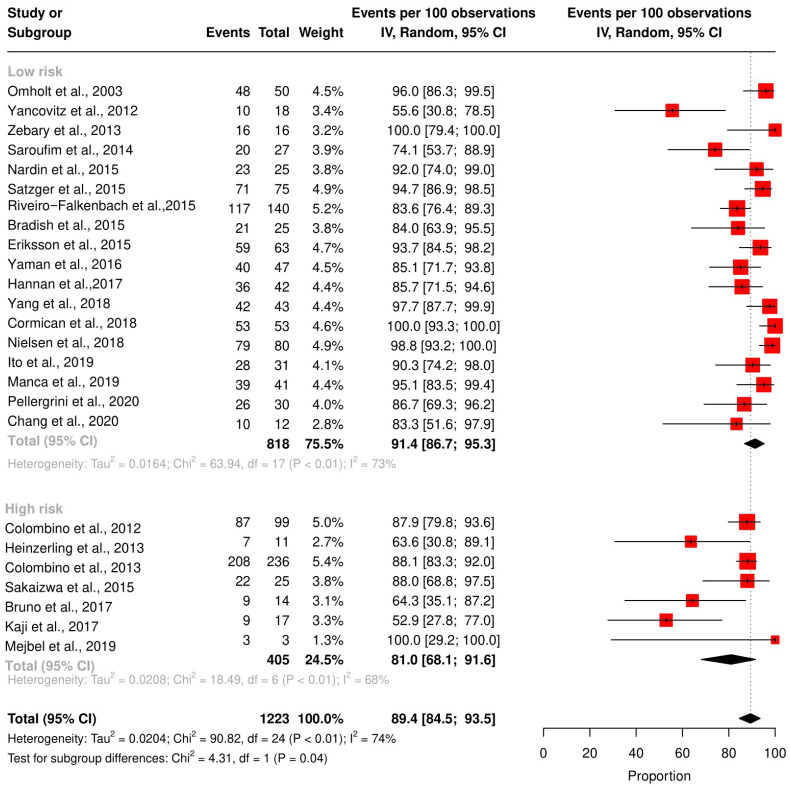
Subgroup analysis of the *BRAF* concordance rate based on QUIPS score. Low-risk bias studies were [27,29,32,33,34,35,36,37,39,40,42,44,45,46,47,48,50,51], high-risk bias studies were [28,30,31,38,41,43,49], red square: point estimate of each study, and black rhomb: summary estimate of each subgroup.

**Figure 6 ijms-24-16281-f006:**
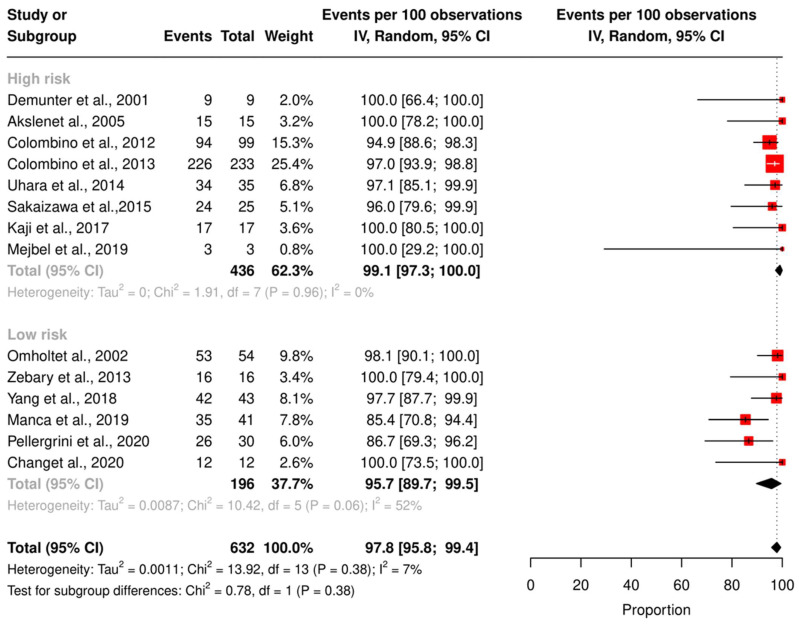
Subgroup analysis of the *NRAS* concordance rate based on QUIPS score. High-risk bias studies were [28,30,38,43,49,52,54,55], low-risk bias studies were [32,45,48,50,51,53], red square: point estimate of each study, and black rhomb: summary estimate of each subgroup.

**Table 1 ijms-24-16281-t001:** Characteristics of the studies included in the meta-analysis for *BRAF* concordance rate.

Study	Country	Technique	Total Cohort (*n*)	Patients with Concordant Status (*n*)	Risk of Bias Score (Study Quality)
Omholt et al., 2003 [27]	Sweden	PCR	50	48	5 (high)
Yancovitz et al., 2012 [29]	USA	MS-PCR	18	10	4 (high)
Colombino et al., 2012 [28]	Italy	ADS	99	87	2.5 (low)
Heinzerling et al., 2013 [31]	Germany	PCR	11	7	1 (low)
Zebary et al., 2013 [32]	Sweden	PCR	16	16	4 (high)
Colombino et al., 2013 [30]	Italy	ADS	236	208	3 (low)
Saroufim et al., 2014 [33]	Lebanon	PCR	27	20	4.5 (high)
Nardin et al., 2015 [36]	France	Pyrosequencing	25	23	6 (high)
Satzger et al., 2015 [39]	Germany	Ultra-deep NGS	75	71	6 (high)
Riveiro-Falkenbach et al., 2015 [37]	Spain	Cobas + IHC	140	117	5 (high)
Bradish et al., 2015 [34]	USA	PCR	25	21	5 (high)
Eriksson et al., 2015 [35]	Sweden	IHC	63	59	4 (high)
Sakaizwa et al., 2015 [38]	Japan	DS	25	22	3 (low)
Yaman et al., 2016 [40]	Turkey	Pyrosequencing + IHC	47	40	5 (high)
Bruno et al., 2017 [41]	Italy	PNA, IHC, capillary seq	14	9	2 (low)
Hannan et al.,2017 [42]	Ireland	PCR and IHC	42	36	5.5 (high)
Kaji et al., 2017 [43]	Japan	MassARRAY	17	9	3.5 (low)
Yang et al., 2018 [45]	USA	IHC and direct + Sanger sequencing	43	42	6 (high)
Cormican et al., 2018 [46]	Ireland	PCR and IHC	53	53	5.5 (high)
Nielsen et al., 2018 [44]	Denmark	Cobas test and IHC	80	79	5.5 (high)
Manca et al., 2019 [48]	Italy	Targeted NGS	41	39	5.5 (high)
Ito et al., 2019 [47]	Japan	IHC	31	28	4 (high)
Mejbel et al., 2019 [49]	USA	NGS	3	3	2.5 (high)
Pellergrini et al., 2020 [51]	Italy	PCR and IHC	30	26	5.5 (high)
Chang et al., 2020 [50]	USA	SNaPshot assays, Sanger sequencing, MS PCR	12	10	4 (high)

ADS: automated direct sequencing, DS: direct sequencing, IHC: immunohistochemistry, MS-PCR: methylation specific PCR, NGS: next-generation sequencing.

**Table 2 ijms-24-16281-t002:** Characteristics of the studies included in the meta-analysis for *NRAS* concordance rate.

Study	Country	Technique	Total Cohort (*n*)	Patients with Concordant Status (*n*)	Risk of Bias Score
Demunter et al., 2001 [52]	Belgium	DOP-PCR	9	9	1
Omholtet al., 2002 [53]	Sweden	PCR and SSCP	54	53	5
Akslenet al., 2005 [54]	Germany	SSCP	15	15	3
Colombino et al., 2012 [28]	Italy	ADS	99	94	2.5
Colombino et al., 2013 [30]	Italy	ADS	233	226	3
Zebary et al., 2013 [32]	Sweden	PCR	16	16	4
Uhara et al., 2014 [55]	Japan	PCR	35	34	2.5
Sakaizawa et al., 2015 [38]	Japan	DS	25	24	3
Kaji et al., 2017 [43]	Japan	Sequenom MelaCarta MassARRAY	17	17	3.5
Yang et al., 2018 [45]	USA	Direct and Sanger sequencing + IHC	43	42	6
Manca et al., 2019 [48]	Italy	Targeted NGS	41	35	5.5
Mejbel et al., 2019 [49]	USA	NGS	3	3	2.5
Pellergrini et al., 2020 [51]	Italy	PCR and IHC	30	26	5.5
Chang et al., 2020 [50]	USA	SNaPshot assays, Sanger sequencing, MS PCR	12	12	4

ADS: automated direct sequencing, DS: direct sequencing, IHC: immunohistochemistry, DOP-PCR: degenerate oligonucleotide-primed polymerase chain reaction, MS PCR: methylation-specific polymerase chain reaction, SSCP: single-strand conformational polymorphism.

**Table 3 ijms-24-16281-t003:** Characteristics of the studies reporting the *c-KIT* status.

Study Name	Country	Technique	Population Cohort (*n*)	Patients with Concordant Status (*n*)	Risk of Bias Score
Zebary et al., 2013 [32]	Sweden	Sequencing	16	16	4
Sakaizawa et al., 2015 [38]	Japan	DS	25	22	3
Mejbel et al., 2019 [49]	USA	PCR	3	3	2.5

DS: direct sequencing, PCR: polymerase chain reaction.

**Table 4 ijms-24-16281-t004:** Reported additional genes for the concordance between the primary melanoma and matched metastasis.

Study	Total Patient	Gene	Mutation	N Mutated Primary	N Mutated Metastasis	N Mutated Samples	N Concordant Patients
Chang et al., 2020 [50]	11	*TERT*	promoter (146 C > T)	4	11 *	15	6
Yang et al., 2018 [45]	41	*TERT*	promoter	N/A	N/A	N/A	28
Kaji et al., 2017 [43]	17	*CDK4*	R24C	1	2	3	0
	17	*KRAS*	G12A	1	0	1	0
	17	*NEK10*	E379K	1	0	1	0
	17	*EPHB6*	G404S	2	1	3	1
Manca et al., 2019 [48]	41	*TP53*	V216M	0	2	2	0
	41	*TP53*	R158C	0	1	1	0
	41	*MAP2K1*	Q46Tter	0	1	1	0
	41	*MAP2K1*	Q110Ter	0	1	1	0
	41	*PTEN*	G127	0	1	1	0
	41	*PTEN*	Q110ter	0	1	1	0
	41	*CCND1*	G103R	0	1	1	0
	41	*CDKN2A*	G23S	1	0	1	0
	41	*CDKN2A*	R131H	1	0	1	0
	41	*PIK3CA*	T1031I	1	0	1	0
	41	*PIK3CA*	G1049S	1	0	1	0
	41	*TP53*	E286K	1	2	3	0
	41	*TP53*	R196L	1	0	1	0
	41	*MAP2K1*	Q383ter	1	0	1	0
	41	*MAP2K1*	Q243Ter	1	0	1	0
	41	*MAP2K1*	Q354ter	1	1	2	0
	41	*RB1*	Q354ter	1	1	2	0
	41	*PTEN*	G165R	1	0	1	0
	41	*CDKN2A*	A40V	1	1	2	1
	41	*PIK3CA*	V344M	1	1	2	1
	41	*TP53*	R196Ter	1	1	2	1
	41	*TP53*	P278L	1	1	2	1
	41	*TP53*	P278S	1	1	2	1
	41	*PIK3CA*	V344A	2	0	2	0
Mejbel et al., 2019 [49]	3	*RAC1*	P29S	0	1	1	0
	3	*CTNNB1*	S37F	1	0	1	0
	3	*HNFA1*	A269T	1	0	1	0
	3	*TP53*	H179Y	1	1 *	2	1

* (patients with multiple metastases).

**Table 5 ijms-24-16281-t005:** Concordance studies by metastatic site.

Study	Lymph Node Metastasis	Brain Metastasis	Visceral Metastasis	Subcutaneous Metastasis	Skin Metastasis/ Other Type
	Concordance Rate % (Concordant Cases/Total Cases)				
Heinzerling et al., 2013 [31]	-	-	-	-	100 (7/7)
Zebary et al., 2013 [32]	100 (15/15)	-	-	-	100 (1/1)
Colombino et al., 2013 [30]	90 (109/120)	92 (22/24)	93 (37/40)	-	77 (40/52)
Saroufim et al., 2014 [33]	89 (16/18)	-	-	67 (4/6)	50 (2/4)
Nardin et al., 2015 [36]	100 (14/14)	-	80 (4/5)	92 (11/12)	
Bradish et al., 2015 [34]	-	50 (2/4)	-	-	92 (11/12)
Yaman et al., 2016 [40]	83 (34/41)	-	-	-	-
Kaji et al., 2017 [43]	53 (9/17)	-	-	-	-
Manca et al., 2019 [48]	94 (17/18)	-	100 (3/3)	-	-
Pellergrini et al., 2020 [51]	89 (17/19)	100 (1/1)	50 (1/2)	-	88 (7/8)
Total samples in all studies	262	29	50	18	84

**Table 6 ijms-24-16281-t006:** *BRAF* concordance rate by gender in the reported studies.

Study	Risk of Bias Score	Total Cohort (*n*)	Total Females (*n*)	Total Males (*n*)	Concordant Female Patients (*n*)	Concordant Male Patients (*n*)
Heinzerling et al., 2013 [31]	1	11	7	5	4	4
Saroufim et al., 2014 [33]	4.5	27	7	19	5	15
Bradish et al., 2015 [34]	5	25	13	11	11	9
Yaman et al., 2016 [40]	5	47	18	29	14	26
Kaji et al., 2017 [43]	3.5	17	9	8	3	5
Total		127	54	72	37	59
Concordance rate % (*p*-value = 0.19)					68.5	81.9

## Data Availability

All data relevant to this study are included in the manuscript.

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
