# Peer review of "Genetic Concordance in Primary Cutaneous Melanoma and Matched Metastasis: A Systematic Review and Meta-Analysis"

_ijms, 2023, doi:10.3390/ijms242216281_

Round 1
Reviewer 1 Report
Comments and Suggestions for Authors
In this systematic review, Kerkour and colleagues evaluated the concordance of mutational status of driver genes in matched primary and secondary melanomas. A high concordance for BRAF and NRAS mutation profiles was reported.
- Introduction appropriately focused on the need to collect data from the mutation analyses on matched samples from the same patients with melanoma.
- The workflow diagram at the basis of the systematic review of the literature is clear.
- Results are rightly presented with inclusion of Tables reporting the techniques used for mutation analysis in each cohort, providing geographical origin and total number of cases analyzed. Table 5 with the comparison of the concordance rates by metastatic sites is a sure added value.
- References are complete and appropriate
- Data are well presented and tables describing all studies are accurately detailed. Conclusions are consistent with the reported data.
Author Response
"Please see the attachment."

Reviewer 2 Report
Comments and Suggestions for Authors
In this study, authors comprehensively analyzed the current knowledge on the concordance of mutated genes between primary CM and their matched metastasis. Also they have analyzed various methods of detection of mutations. they should include data on NF1 mutations.
Introduction - is fine and explains rationale for study.
Results - explain the metanalysis quite well.
Discussion - elaborates current status and highlights current study findings with it quite well.
Appropriate recent references are cited for every section.
In all tables - they should refer to the study by appropriate reference no. as well - this will be more apt for journal referencing style.
A figure should be added summarizing their results for better visualization of the findings. Quality, resolution of Figure 1 needs to be improved.
Author Response
"Please see the attachment."

Reviewer 3 Report
Comments and Suggestions for Authors
Kerkour et al.'s study entitled "Genetic Concordance in Primary Cutaneous Melanoma and Matched Metastasis: A Systematic Review and Meta-Analysis" systematically reviewed all literatures on primary melanoma and its matched metastasis. Their findings show that there is significant variation in the genes implicated in the genetic makeup of melanoma. The study's rationale is robust, and it addressed a valid question of the correlation between the mutation status of primary CM and matching metastases for multiple cancer genes. The study is well-written, and the design is excellent. The method is thoroughly described. Overall, the study made a reasonable point that molecular testing in metastatic tissue is a more acceptable way for detecting BRAF status in order to adapt treatment decisions.
The manuscript has only one minor comment:
1. Throughout the text, the author utilized the BRAF V600 mutation, which is not the right annotation of mutation. The author must use BRAF V600E instead of BRAF V600.
Author Response
"Please see the attachment."

Reviewer 4 Report
Comments and Suggestions for Authors
The main question addressed by the research is relevant and interesting, particularly for clinical and treatment decision purposes, as understanding the genetic concordance or discordance between primary and metastatic tumors is crucial in determining appropriate therapies for melanoma patients.
In terms of originality, the research contributes to the existing body of knowledge by providing updated data and analysis.
Regarding the writing quality, the text appears to be well-written and structured in a logical manner. The ideas are presented clearly and are relatively easy to read and understand.
The conclusions drawn in the text address the primary question posed in the research, which is to determine the level of concordance between primary and metastatic melanoma tumors. Therefore, the conclusions align with the main research question.
However, I have several comments, which are described below.
1. In lines 42-43, 77-80, 82-86, and so on, the citations must be at the end of the sentence.
2. In the text, the indications for all the figures are inconsistently inserted. These should be corrected to consistent ones.
3. In all the Tables, I think it would be appropriate to introduce a column that correlates the articles with the citations in the bibliography.
4. As for the conclusion section, it does not specifically outline specific future research directions or suggest potential areas for further investigation.
Comments on the Quality of English Language
Minor editing of English language required.
Author Response
"Please see the attachment."
